# Marital rape and its impact on the mental health of women in India: A systematic review

Nandini Agarwal[1]*, Salma M. Abdalla[2], Gregory H. Cohen[2]

**1** Boston University School of Public Health, Boston, Massachusetts, United States of America,
**2** Epidemiology Department, Boston University School of Public Health, Boston, Massachusetts, United States of America

* nanagrwl@bu.edu

## Abstract

This systematic review aims to describe the prevalence of marital rape in India, the analytic methods employed in its study, and its implications on mental health of victims. Online databases, PubMed, Embase, Web of Science and APA Psych, were systematically searched for articles published up until November 2020. Selected articles included those published from or studies conducted in India where the primary exposure was marital rape. The primary outcomes of interest are Post Traumatic Stress Disorder (PTSD) and Depression. Secondary outcomes related to PTSD and depression (e.g., suicidality) included in identified studies were also described. 11 studies were included after excluding studies based on our selection criteria: 9 quantitative studies and 2 qualitative studies. Sexual coercion by intimate partner was highly prevalent, ranging from 9%-80% and marital rape ranged from 2%-56%. Many of the studies reported statistically significant associations between marital rape and mental health outcomes, including clinical depression (7 of 8); PTSD (1 of 3). Quantitative studies were assessed for quality and risk of bias using the NIH Quality Assessment Scale and the modified Newcastle Ottawa Scale for cross-sectional and observational cohort studies, and most exhibited a low risk of bias. Qualitative studies identified a broad range of exposures and psychological sequlae of marital rape not captured by quantitative studies. Included publications exhibit a low to moderate association between marital rape and adverse mental health outcomes. Qualitative data also supplements these findings and provide relevant context. Further research on marital rape, its prevalence and consequences, is needed to advance policy, and health infrastructure on the subject.

## Introduction

Marital rape is defined as non-consensual sexual intercourse with one's spouse [1]. Acts of forced sexual contact, including but not limited to vaginal penetration are considered marital rape in India, with the condition that the wife is younger than fifteen years of age [2]. Sexual violence in marital relationships is one of the most privatized and least addressed forms of violence [1]. In India, while rape outside of marriage is a crime, rape within marriage is not

**Data Availability Statement:** Yes - all data are fully available without restriction. All relevant data are within the manuscript and its Supporting Information files.

**Funding:** The author(s) received no specific funding for this work.

**Competing interests:** The authors have declared that no competing interests exist.

necessarily considered criminal and is socially tolerated, as outlined in Box 1 [3, 4]. Underreporting of sexual Intimate Partner Violence (IPV) masks the true burden of sexual domestic violence [5, 6]. Most sexual violence in India occurs within marriages but it is estimated that only about 10% of victims report spousal sexual abuse [7]. This suggests that the lack of accurate reporting on marital rape, has not only undermined an appreciation for the true burden of sexual assault in Indian society, but has also contributed to the paucity of research on the psychological impact of spousal sexual abuse. Further, women who are victims of spousal sexual abuse often suffer from other types of IPV as well; physical, emotional, and psychological, thus bearing a particularly potent burden of exposure and psychiatric risk.

## Box [1]. Legal and Social Status of Marital rape in India

### Legal and Social Status of Marital Rape in India

Sexual intercourse between a husband and his wife is not considered rape if she is over 15 years of age, per the second exemption in Section 375 of the Indian Penal Code. ADDIN CSL_CITATION {"citationItems":[{"id":"ITEM-1","itemData":{"URL":"https://devgan.in/ipc/section/375/","accessed":{"date-parts":[["2021","2","27"]]},"id":"ITEM-1","issued":{"date-parts":[["0"]]},"title":"IPC Section 375 - Rape |

Despite the evidence of adverse effects of marital rape on the mental and emotional health of victims, in India, rape in the context of marriage remains largely unaddressed in clinical practice, scientific research, and public health surveillance [8–10].

Previous studies have demonstrated associations between rape and several mental health outcomes, including depression, Post-Traumatic Stress Disorder (PTSD), and sleep disorders [11, 12]. Survivors of both physical and sexual IPV have recounted suffering from adverse mental health outcomes [13, 14]. Countries that criminalize marital rape, such as the United States also report marital rape victims suffering from adverse mental health outcomes such as PTSD and depression [15] with women raped by their spouses exhibiting even higher rates of anger, depression, and suicidal feelings as compared to those assaulted by strangers [10, 16]. Martial rape also impacts help-seeking behavior among survivors, which may vary across cultures. About 61% victims reported seeking help for sexual IPV in New Zealand, 40% in Tanzania and Jordan but only 24–26% in India; with only 2–4% seeking help from authorities in India [17]. This inability to talk about the abuse and seek help also negatively impacts women's mental health, exacerbating the stress, anxiety and depressive symptoms experienced by victims [17]. This evidence from other countries suggests that the mental health burden from marital rape in India is potentially high. However, societal inattention to the issue, lack of infrastructure and procedure for screening, comorbid exposures and lack of help-seeking avenues, it has been challenging to estimate the consequences of marital rape within the Indian society.

This paper attempts to address this gap and map out the literature on the mental health consequences of marital rape and spousal sexual abuse within the Indian community. This systematic review will address the question; does marital rape result in adverse mental health outcomes such as depression, PTSD, anxiety disorders, and somatic symptoms?

## Methodology

### Search strategy

We conducted a systematic database search from January 1945 to November 2020, including compiling initial list of articles and then excluding non-relevant, duplicate, and ineligible papers. The search included 4 databases, namely PubMed, Embase, Web of Science and APA Psych Info in our search along with snowball selection from references of relevant articles. The key search terms were "marital rape", "spousal violence", "intimate partner violence", "domestic violence", "married women", "India", "Indian", "Southeast Asia", "depression", "PTSD" and "mental health". To keep the search broad, no filters were applied. No time or language restriction was applied to cover any and all literature published, including grey literature so that it would be included in the search. Articles published up until November 2020 were included. A process flow chart according to the PRISMA standard has been charted in Fig 1.

All articles were uploaded to Rayyan QCRI for review. All three authors had access to the Rayyan account. The initial round of review involved title and abstract selection was performed by NA. After exclusion of duplicates and a bulk of non-relevant articles, all three

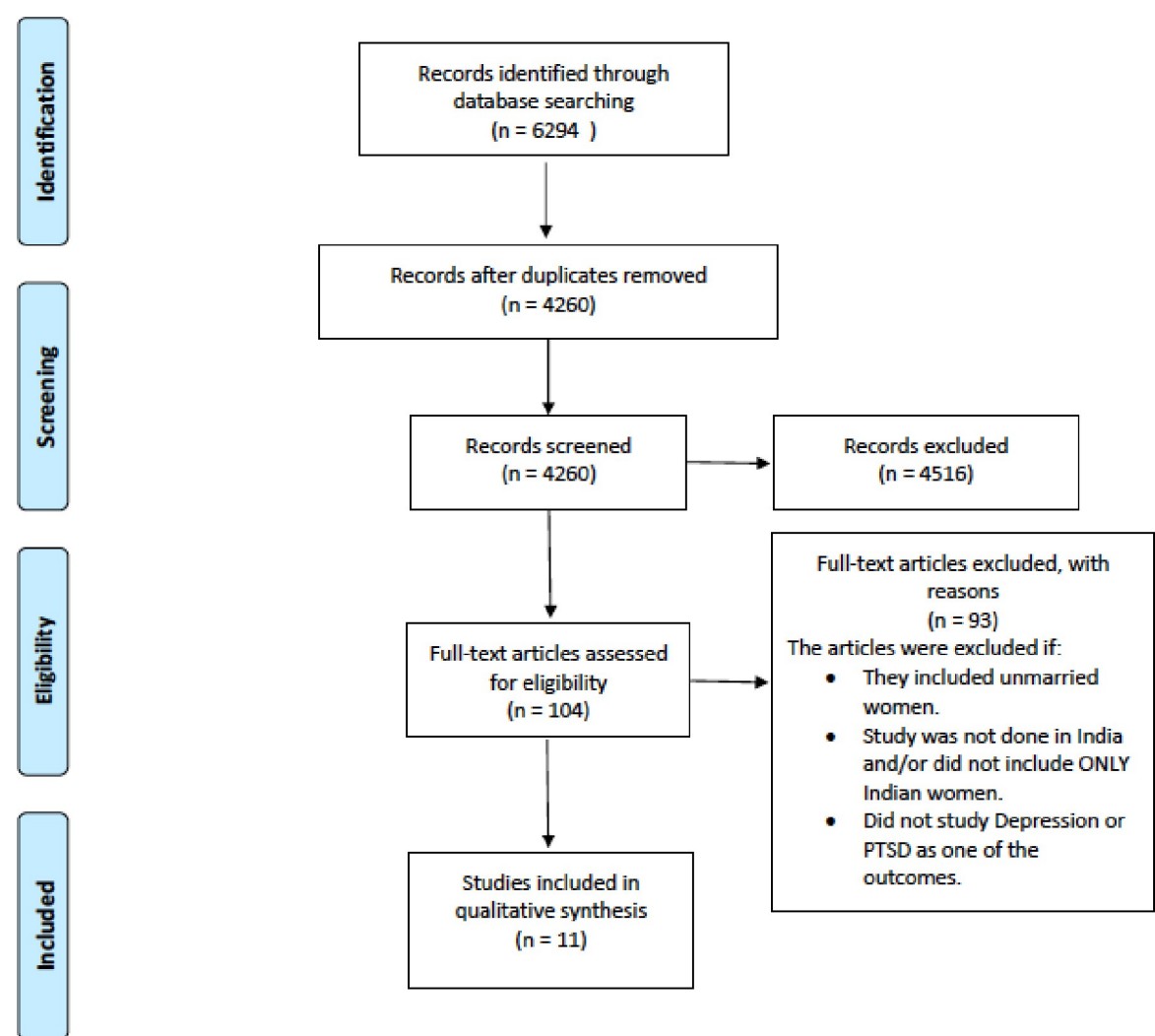

**Fig 1. PRISMA flow diagram for systematic review.**

reviewers performed study selection based on a predetermined criterion using full text search for the remaining 104 studies. For the purpose of this systematic review no filters were applied pertaining to study design. Both qualitative and quantitative studies, along with reviews were included in our search to identify the depth and extent of research done regarding prevalence of marital rape experienced by Indian women and the effects on mental health.

### Study selection criteria

We selected studies that included married Indian women who were exposed to spousal sexual abuse, and whose mental health outcomes were studied. No filters were applied in terms of summary measures used. We excluded studies that did not include spousal sexual abuse, only studied non-penetrative sexual abuse, or that focused on sexual abuse experienced by married Indian women that was not perpetuated by a spouse. Studies that included unmarried women were excluded.

If associations between marital rape and mental health outcomes was not explored, then those studies were excluded as well. Mental health outcomes included Major Depressive Disorder (MDD) and Post Traumatic Stress Disorder (PTSD). If studies measured one or both of these outcomes, they were included. Other adverse mental health outcomes (like suidcidal ideation and attempts) measured by these studies were reported as secondary outcomes.

The sample for potential studies should be of married, Indian women. Any other nationality, if exclusively mentioned would not be considered. Any study done with Indian women in any setting other than India will not be considered. Studies with the South Asian community in United States of America, Europe or any other setting were excluded.

### Ethics statement

This study did not involve human participants or animal subjects.

### Patient involvement

No patients were involved nor was any patient input included at any stage in this paper.

### Results

Eleven studies were included after excluding studies based on our selection criteria: nine quantitative studies and two qualitative studies. One study was excluded because of unavailability of full text, while another was excluded due to the inclusion of both married and unmarried women in the sample. One study was excluded because it did not assess presence of sexual violence at the start of the study period but only checked for its presence during follow-up, while two were excluded because the outcomes did not include either depression or PTSD. All studies included in this review are summarized in four tables. Tables 1 and 2 provide a short study description including the study objective, sampling strategy and size, recruitment strategy employed, a brief description of the study population and the study design.

Tables 3 and 4 provide a brief description of types of intimate partner violence experienced by participants, mental health outcomes studied, including the tools used to measure the prevalence of both exposure and outcome variables. The analytic methods employed and the results of each of these studies are also included in the following tables.

Of the eleven studies, eight studies assessed the association between spousal violence (physical, sexual and psychological) and clinical depression [18, 20–23, 27–29], while two studied antenatal and postnatal depression as outcomes [23, 24]. Only four studies explored the

**Table 1. Quantitative study description.**

| Study | Objective | Sample type and size | Study design | Sampling strategy | Recruitment strategy | Study Population |
|---|---|---|---|---|---|---|
| [18] | To describe the prevalence and nature of PTSD among married women who report IPV, and examine relationships between sexual coercion and PTSD as well as between PTSD and depression | Clinical (N = 105) | Cohort study | Facility-based convenience sampling | Consecutive women presenting at the psychistric unit of National Institute of Mental Health and Neuro Sciences in South India recruited (2003–2005) if they met inclusion criteria | Women who visited the outpatient psychiatric department of National Institute of Mental Health and Neurosciences in South India. |
| [19] | To study the hypothesis that domestic violence is an independent risk factor for a variety of adverse health conditions | Population level (N = 1750) | Cohort study | Facility-based simple random sampling | Subjects randomly selecetd and were recruited via letters followed by home visits. | Married women living in the catchment area of a primary healthcare center in Goa were assessed twice after recruitment between the years 2001–2004. |
| [20] | To study the prevalence of domestic violence among spouses of alcohol-dependent men and prevalence of common mental disorders among these women | Clinical (N = 60) | Cross-sectional study | Facility-based convenience sampling | Consecutive women accompanying their spouses attending the de-addiction clinic recruited if they met inclusion criteria | Spouses of alcohol-dependent males who attended a de-addiction center in South Kerala, India. |
| [21] | A prospective observational study designed to assess the prevalence of IPV during pregnancy and its effect on maternal and neonatal health | Clinical (N = 400) | Cohort study | Facility-based purposive sampling | Pregnant women attending the outpatient clinic recruited if they met the inclusion criteria. | Pregnant women, between 20–28 weeks of gestation who visited the outpatient department at a tertiary care hospital in Delhi, India were followed until delivery in the period December 2013- April 2015. |
| [22] | To determine the prevalence of lifetime domestic violence and its association with mental and physical health of economically disadvantaged married women | Clinical (N = 219) | Cross-sectional study | Facility-based convenience sampling | Women (aged 18 years or older), spoke and read Gujarati, and were seeking services at a community health center recruited. | Married women aged between 18–62 years who utilized community health services in Rajkot city. |
| [23] | To estimate the prevalence of prenatal depression and associated risk factors among pregnant women in Bangalore | Clinical (N = 280) | Cross-sectional study | Antenatal Clinic (ANC)-based purposive sampling | Consecutive pregnant women attending the antenatal clinic at Jaya Nagar General Hospital (also known as Sanjay Gandhi Hospital), a public sector hospital in Bangalore, India. | Pregnant women who attended the antenatal clinic in Jaya Nagar General Hospital (Sanjay Gandhi Hospital) in Bangalore. |
| [24] | A cross-sectional study nested within a prospective cohort study designed to assess the prevalence and risk factors for suicidal ideation, attempts and/or behavior among married pregnant women in South India | Clinical (N = 462) | Cross-sectional study | ANC-based purposive sampling | Pregnant women who attended the Antenatal clinic at the Government Referral Hospital (GRH) in South Bangalore, India were recruited if they were in between 5–20 weeks of pregnancy. | Pregnant women who registered with the Antenatal clinic at the Government Referral Hospital (GRH) in South Bangalore, India. |
| [25] | To study the effect of the patriarchal structure of the Indian society's impact on prevalence of IPV, coping mechanisms and its psychological impact on women | Clinical (N = 64) | Cross-sectional study | Simple random sampling | Flyers advertiisng the study were placed in public locations (urban and rural) and women who responded and subsequently met the study criteria were recruited. | Currently married women, at least 18 years of age (18–56 years) were recruited from the Indian state of Tamil Nadu. The marriage was the first for both husband and wife |
| [26] | To study the association of IPV during pregnancy and adverse mental health outcomes, especially depression and PTSD | Clinical (N = 203) | Cross-sectional study | ANC-based purposive sampling | Consecutive admissions to the antenatal outpatient clinic of a large urban obstetric centre in southern India over a two month period were recruited. | Currently married and pregnant women, aged 18–4 years attending an outpatient antenatal clinic in a public hospital in Bangalore, South India. |

association between spousal violence and Post Traumatic Stress Disorder (PTSD) [18, 25–27]. Two studies [18, 25] assessed for suicidality including ideation and attempts. Only one study [28] looked at Idioms of Distress as listed in the DSM-V criteria. See Fig 2 for outcome distribution in included studies.

## Quality of studies

The included quantitative studies were assessed for quality and risk of bias using the NIH Quality Assessment Scale and the modified Newcastle Ottawa Scale for cross-sectional and observational cohort studies, respectively [30, 31].

Most studies exhibit low risk of bias in terms of comparability due to adjustment of appropriate confounders (Table 5). It is notable that the study conducted by Chowdhary et.al. (2008) scored high on the quality rating as well had low risk for bias.

## Type of spousal violence

The forms of spousal violence that were studied included physical, sexual, verbal, psychological and emotional violence. Sexual violence was common to all the studies included (Fig 3). Some studies did not study sexual abuse as a separate exposure but combined all forms of violence into one variable, domestic violence [19, 26].

Some studies (n = 6) considered the combined effect of spousal violence on mental health outcomes [20–24, 32]. None of the studies exclusively examined the effects of sexual violence, including rape, coercion, forced prostitution, forced to watch, and enact pornographic material or other forms of non-penetrative sexual violence on depression. Only one study [18] assessed the causal relation between sexual coercion (including marital rape) and PTSD; while only one study [26] that enrolled pregnant women and aimed to determine if sexual coercion during pregnancy was related to development of adverse mental health outcomes.

## Prevalence of spousal sexual violence and marital rape

Studies measured domestic violence including physical, sexual and psychological violence. Prevalence of current domestic violence reported by participants ranged from 13%-68% [17,

**Table 2. Qualitative study description.**

| Study | Objective | Study population | Study design | Sampling startegy and sample size | Recruitment strategy | District category | Methodology |
|---|---|---|---|---|---|---|---|
| [27] | To study the impact of domestic violence on mental health in a rural, marginalized community in North India | Women aged 18–80 years who had been or currently were victims of domestic violence and utilized services of either 1 of the 2 partner NGOs were recruited via purposive sampling. The study was conducted in Kangra district in Himachal Pradesh. | Cross-sectional study | Purposive sampling (N = 23) | Participants recruited through the Nishtha health clinic, Mishtha's mobile outreach clinic or at Jagori Nari Adalats (women's courts)by approaching women who were known to have experieneced domestic violence by women acitivists employed at Nishtha. | Rural | 23 semi-structured interviews conducted in the presence of an expert interpreter for Hindi who was not part of the community. The interview was based on the World Health Organization Multi-Country study on Women's Health and Domestic Violence, after adjusting for local cultural practices and views. Participant observation was also attempted to obtain a deeper understanding of the women's beliefs, views and reactions to this sensitive subject. |
| [28] | To study the impact of domestic violence on female survivors using the concept of Idioms of Distress (IOD) | Survivors of domestic violence were interviewed. These participants were contacted with the help of an NGO in Bengaluru. | Cross-sectional study | Purposive sampling (N = 6) | Women who sought help at an NGO for domestic violence in Bangalore were approached and the first 6 women who agreed to aprticipante were recruited. | Urban | 6 semi-structured interviews were conducted. The interview questions were based on previous literature and culturally appropriated by taking the experiences of experts working at the NGO into account. |

**Table 3. Study findings in terms of effect of intimate partner violence on mental health outcomes (Quantitative).**

| Study | Spousal violence | Tools used to establish presence of domestic violence | Mental health outcomes/ psychological morbidities | Tools used for psychological assessment | Statistical measures | Results |
|---|---|---|---|---|---|---|
| [18] | Physical, sexual, verbal and emotional violence | 1) Index of Spouse Abuse (ISA) to measure physical abuse (ISA-P) and non-physical abuse (ISA-NP). 2) Sexual Experiences Scale (SES) to measure sexual coercion | 1) Depression 2) Post-Traumatic Stress Disorder (PTSD) | 1) Beck Depression Inventory (BSI) 2) Post-Traumatic Symptom Checklist (PCL) | Frequency analysis for distribution of socio-demographic characteristics and prevalence rates. Chi-square tests and independent sample t-test done to compare characteristics of women who experienced IPV to those who did not. Partial correlation done to test association between violence severity and PTSD, while controlling for depression. | 59 out of the recruited 105 women reported presence of IPV. Out of these women, 41 or 70% reported being coerced into sexual activities by their spouse. Of the women suffering from spousal abuse, 12% met the criteria for PTSD and 99% met the criteria for depression. Despite few women meeting the PCL criteria for PTSD, most participants exhibited signs of subthreshold PTSD, which can be just as debilitating as clinically diagnosable PSTD and help for which is rarely sought. 42% of the women were found to be suffering from severe depression. PTSD severity and sexual coercion were seen to be significantly correlated (r = 0.39), while depression and sexual coercion were not significantly correlated (r = 0.06). |
| [19] | Physical, sexual and verbal violence | Structured interview to assess domestic violence | 1) Depression 2) Suicidal behavior | The Revised Clinical Interview Schedule (CIS-R) used PROQSY software to generate an ICD-10 classification for depressive disorder. | Univariate analyses were carried out to assess the association between each type of spousal violence and each health outcome. Logistic regression to adjust for confounders in testing association. | Out of 1750 women, 230 (13%) reported experiencing one or more forms of spousal violence within the last 3 months (current violence). Sexual violence was the least reported form of violence, which can be attributed to stigma or the highly specific definition of sexual violence incorporated in this study. Forced sexual intercourse by spouse alone was counted as sexual abuse. All types of violence were significantly associated with depression, PTSD and attempted suicide. However, only STIs and attempted suicide displayed a statistically significant relation with spousal violence in the longitudinal analysis. |

*(Continued)*

**Table 3.** (Continued)

| Study | Spousal violence | Tools used to establish presence of domestic violence | Mental health outcomes/ psychological morbidities | Tools used for psychological assessment | Statistical measures | Results |
|---|---|---|---|---|---|---|
| [20] | Physical, psychological and sexual violence by spouse and/or family members | Domestic Violence Questionnaire (DVQ) | 1) Depressive disorders | Diagnosed using DSM-IV-TR criteria, using MINI Malayalam Version | Frequency analysis for distribution of socio-demographic characteristics and prevalence rates. Association assessed using Chi-square test or Fisher's exact test as appropriate. Pearson's product moment correlation coefficient and point bi-serial correlation coefficient used to assess correlation between variables for continuous and dichotomous variables, respectively. | 41 (68.3%) women reported having experienced domestic violence, with the different types of violence, psychological, physical and sexual combined together as 1 variable. Major depressive disorder (MDD) was diagnosed in 15 women, 5 (8.3%) currently suffering from it, 6 (10%) had recurrent MDD and 4 (6.7%) had been diagnosed in the past. |
| [21] | Physical, sexual and emotional violence | Violence Assessment Screen | Clinical Depression | ICD-10 classification | Frequency analysis for distribution of socio-demographic characteristics and prevalence rates. Associations assessed using independent sample t-test and Pearson Chi-square test (Fisher's exact test when appliable). | 49 out of 400 women experienced IPV, while only 7 (1.8%) reported having experienced sexual violence. Women who reported sexual violence also suffered from both emotional and physical violence. The spouses of these women suffered from some substance use disorder (alcohol, tobacco or smoking). Of all the women who reported IPV, 19 (46.3%) were diagnosed with major depressive disorder. |
| [22] | Physical, sexual and emotional violence | A questionnaire from a study of IPV among Asian India, Pakistani and Filipino women in USA (permission obtained) | Depression | 1) Perceived health status using the Duke Health Profile (DUKE)- Includes health measures (physical, social, mental, general, perceived health and self-esteem) and dysfunctional measures (depression, anxiety, pain and disability) 2) Patient Health Questionnaire (PHQ-9) | Frequency analysis for distribution of socio-demographic characteristics and prevalence rates. Pearson Chi-square and independent samples t-test used to compare particiapnts who experience IPV with those who did not. Logistic regression and multiple linear regression to test associations. Multiple linear regression analysis also conducted to test the association between health and dysfunction. | 83 women reported prevalence of IPV, 40% of which experienced only physical abuse. 30% of the women experienced all 3 forms of abuse. Women who reported sexual abuse experienced forced attempts to have sex (28/83), actually having sex i.e., marital rape (24/83) and forced to have sex with others by husband (13/83). Women with a history of IPV exhibited higher levels of anxiety and depression (DUKE and PHQ-9). The association between spousal IPV and depression was clinically significant (DUKE depression; b = 16.19, p<0.01, PHQ-9 depression; b = 4, p<0.01). |

*(Continued)*

**Table 3.** (Continued)

| Study | Spousal violence | Tools used to establish presence of domestic violence | Mental health outcomes/ psychological morbidities | Tools used for psychological assessment | Statistical measures | Results |
|---|---|---|---|---|---|---|
| [23] | Physical, sexual and emotional violence | 1) The Revised Dyadic Adjustment Scale to measure psychological and emotional abuse 2) Modified Conflicts Tactics Scale to measure spousal physical and sexual violence | 1) Antenatal depression | 1) Edinburgh Postnatal Depression Scale (EPDS) | Descriptive statistics used to summarize the socio-demographic data. Bivariate analysis used to test association. The significant variables (p<0.2) were modeled using multivariate logistic regression to account for confounding. | 100 out of 280 (35.7%) women were diagnosed with prenatal depression as per their EPDS score (mean = 10.61 +- 7.48). Of these 100 women, 11 (11.0%) reported spousal physical and sexual violence. Presence of spousal abuse indicated a five times increased likelihood of pregnant women suffering from depression (cOR = 5.438, 95% CI = 1.6,17.5 and aOR = 5.916, 95% CI = 1.7, 20.5). |
| [24] | Physical, psychological and sexual violence by spouse and/or family members | ICMR Task Force abuse assessment questionnaire | 1) Antepartum and Postpartum depression. 2) Suicidality: Ideation, planning, attempts, method used and reasons for it during lifetime and current pregnancy. | 1) Edinburgh Postnatal Depression Scale-Item 10 (EPDS-Item 10) 2) Suicidal Behaviors Questionnaire- Revised (SBQ-R): Modeled into an 8-part item. | Descriptive statistics used to summarize the socio-demographic data. The independent t-test and ANOVA were employed to compare groups. Fisher's exact test used to examine the association between categorical variables. Pearson correlation coefficient and Spearman's correlation coefficient used to examine the relationship between continuous variables. Multivariate logistic regression used to examine the predictors of suicidal ideation in pregnancy. | 56 (12%) women reported having experienced some form of domestic violence. Lifetime prevalence of suicidal ideation was reported as 11.9%, while it was estimated to be 7.6% during pregnancy. 40% of the 35 women who reported suicidal ideation during current pregnancy had a history of the same. 15 (42.9%) women reported having experienced spousal sexual violence. |
| [25] | Physical, sexual, verbal and emotional violence | 1) Relationship Values Misperceptions Survey (RMS)- Culturally relevant survey designed by authors to measure physical, verbal, psychological and sexual violence perpetuated by husband and instances of mutual physical altercations. 2) The Revised Conflicts Tactics Scale (CTS2) | Post-Traumatic Stress Disorder (PTSD) | Post-Traumatic Stress Diagnostic Scale (PSDS)- Diagnosed as per DSM-IV criteria | Demogrphics assessed using frequency and basic correlations. Bivariate correlation and Multiple Analysis of Variance Assessment run to determine the relationship between independent and dependent variables. | 63 of 64 women (98.43%) reported experiencing some form of domestic violence, while 58% of the women had no idea that they were victims of spousal abuse. 84% women exhibited signs of clinically diagnosable PTSD or Acute Stress Disorder. The types of violence were not correlated with PTSD severity. |

(*Continued*)

**Table 3.** (Continued)

| Study | Spousal violence | Tools used to establish presence of domestic violence | Mental health outcomes/ psychological morbidities | Tools used for psychological assessment | Statistical measures | Results |
|---|---|---|---|---|---|---|
| [26] | Physical, sexual and psychological abuse. Also includes sexual coercion. | 1) Index of Spouse Abuse (ISA) to measure physical abuse (ISA-P) and non-physical abuse (ISA-NP). 2) Sexual Experiences Scale (SES) to measure sexual coercion | 1) Depression 2) Post-Traumatic Stress Disorder (PTSD) | 1) Beck Depression Inventory (BDI) 2) Post-Traumatic Symptom Checklist (PCL) | Chi-square tests used to examine the association between sociodemographic variables and presence of abuse. Students' t tests used to examine the relationship between IPV and severity of somatic symptoms, depression, life satisfaction, and PTSD scores. Logistic regression analysis used to identify the significant predictors of violence. Pearson's correlation coefficients calculated to estimate the strength of association. | 10 (5%) women reported infidelity by spouse (with multiple partners) during pregnancy. Severe sexual coercion during pregnancy was reported by 18 (9%) women as per the SES. Women who were victims of sexual coercion had higher depressive symptoms and poorer quality of life as compared to women who did not experience the same. PTSD scores were also high but not significantly different from the women without a history of sexual coercion by spouse. |

19, 22]. Studies also found that participants reported spousal abuse during pregnancy (12.3%) including emotional (11%), physical (10%) and sexual violence (1.8%) [21, 22, 33]. Sexual coercion by intimate partner was highly prevalent among women included in the studies, ranging from 9%-80% [18, 25, 26]. Despite sexual violence being the least reported out of all other types of spousal abuse, marital rape ranged from 2%-56% [18, 21, 23].

## Mental health outcomes

Adverse mental health outcomes of marital rape were studied. Depression, including antenatal and postnatal depression, and Post Traumatic Stress Disorder (PTSD) were the primary study outcomes. We incidentally came upon some results on suicidality as they were measured as secondary outcomes in included studies. Depression was the most common outcome among the included studies (Table 6).

All the studies controlled for previous diagnosis of mental disorders and patients diagnosed with psychotic disorders but did not control for presence of abuse by anybody other than the spouse. One study also controlled for prevalence of substance abuse disorders among the women [24]. Three studies looked at the relationship between domestic violence perpetuated by the spouse during pregnancy with antepartum and postpartum depression being the outcome [21, 23, 24]. These studies did not control for pregnancy related anxiety or abuse by in-laws or a history of sexual abuse.

**Depression.**   Of the 8 studies that examined depression as an outcome 2 were rated 'good' on the NIH quality assessment tool while 4 were rated 'poor'. One of the studies that was rated good, used population-based sampling, and controlled for a variety of confounders such as spousal substance abuse, marital discord, social and familial support, and a history of mental illness. Even after controlling for confounders the authors found a strong association between all types of spousal violence and Major Depressive Disorder (MDD) and suicidality [19]. Four studies, three rated poor and one rated fair on the NIH Quality Assessment Scale, found

**Table 4. Study findings in terms of effect of intimate partner violence on mental health outcomes (Qualitative).**

| Study | Types of domestic violence | Mental health problems studied | Statistical measures | Results |
|---|---|---|---|---|
| [27] | Physical assault including wife battering, dowry harassment and extreme verbal and emotional violence. Sexual abuse such as forced oral sex, forcibly viewing pornography, threats of forced prostitution, sexual harassment, rape and incest. | Depression, PTSD, and suicidality were observed. These symptoms were self-reported. | Grounded theory was used for systematic data interpretation. Selective coding was followed by clustering codes into categories. Core category identification led to theoretical coding to analyse axes and categories. | Sexual violence was reported by several women, including forced oral sex, threats and even forced prostitution (as reported by 1 woman), sexual harassment in their natal and marital home, vulgar language, marital rape and incest was reported. All women felt that domestic violence affected their health considerably. About 15 women exhibited signs of severe depression, however, only 2 actually mentioned the term. Most women were not familiar with what depression means and categorized their distress as "tension". 13 women specifically used the term "tension" to describe severe mental distress. All women displayed symptoms of PTSD. Severe PTSD was diagnosed in 4 women. 8 participants were diagnosed with sleeping disorders and reported that they were afraid of being sexually assaulted by their partner while sleeping. Suicidal ideation was reported by 18 participants, with 8 women having attempted suicide once or more. 1 woman even attempted a murder-suicide after unwanted sexual coercion and touching of her daughter and herself by her spouse. |
| [28] | Domestic violence themes were categorized as per The Protection of Women from Domestic Violence Act, 2005. Participants were assessed for Physical, Sexual, Verbal, Financial and Emotional violence. | Idioms of Distress (IODs) listed in the DSM-V criteria were explored. Most women exhibited signs of depression such as loss of interest in performing daily activities, sadness, irritability, feeling helpless, and suicidal ideation. | Thematic network analysis which identiifed one global theme, 4 organizing and 19 basic themes. | The authors studied several Idioms of Distress (IODs) including psychological and behavioral idioms. All women had low self-confidence, anxiety, suicidal thoughts, changed goals and aspirations, distrust of people, sleeping and concentration problems. Participants self-reported feelings of sadness and "depression" after suffering from spousal abuse. Suicidal ideation was expressed by none of the women attempted it. Concentration problems impaired their ability to perform daily household tasks, prompting further violence and their careers also suffered. Participants reporting feeling withdrawn, uncontrolled crying, lack of interest and attention, and irritability towards their children. The authors did not study the association of different types of violence with psychological and behavioral idioms. |

statistically significant correlations between depression and IPV [22–24, 26]. Spousal sexual abuse increased the likelihood of diagnosed depression symptoms (as per the ICD-10 classification) [22–24, 26]. Depression and spousal physical (r = 0.04) and non-physical abuse (r = 0.15), as well as sexual coercion (r = 0.06) was not significant [18].

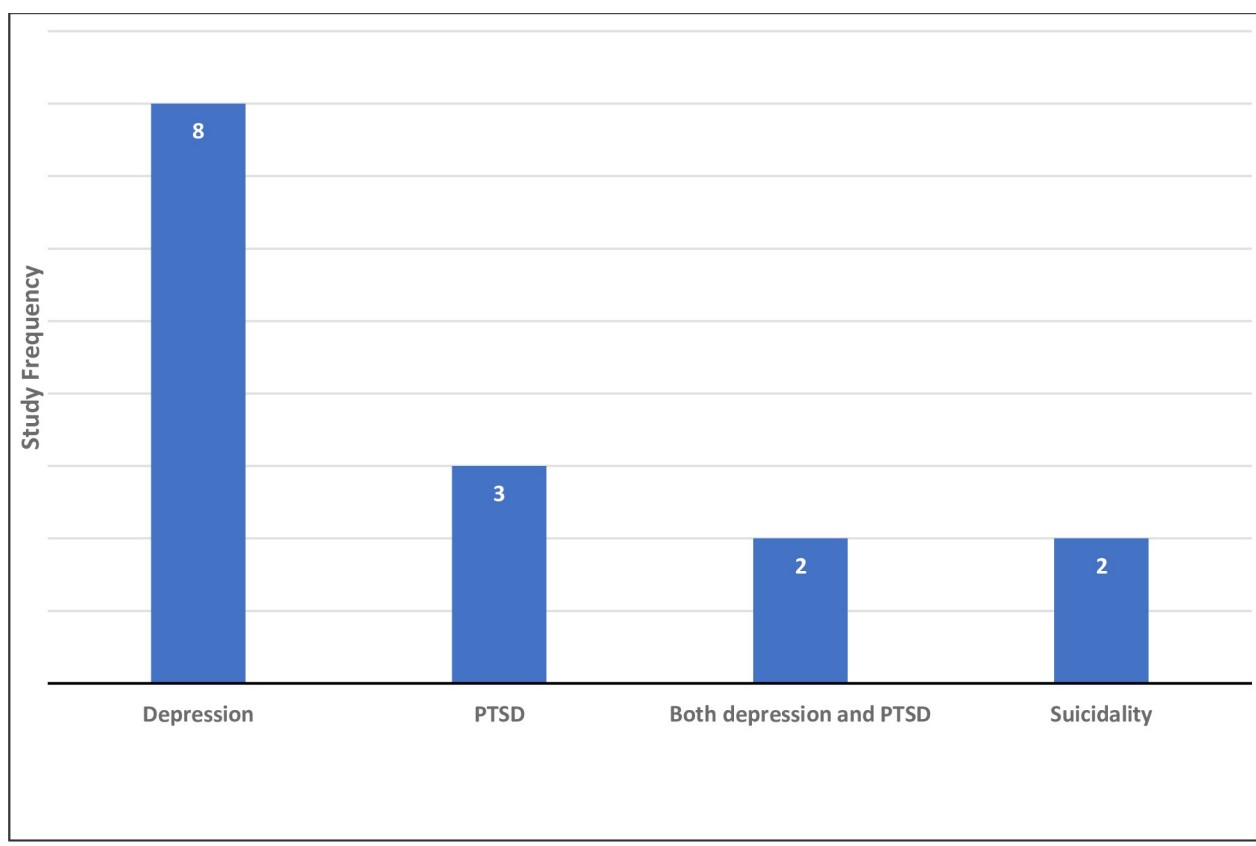

**Fig 2. Distribution of studies based on the outcome.**

**PTSD.** All the studies examining the association between marital rape and PTSD were rated poor according to the NIH Quality Assessment Scale. Among these 3 studies, 2 studies found that sexual coercion and abuse were not statistically significantly associated with PTSD symptoms [25, 26]. However, one of them did observe a significant association between all identification of spousal abuse as a societal problem and help seeking to be significantly associated with chronic PTSD [25]. The third study found a positive, significant correlation between sexual coercion and PTSD severity (r = 0.39) [18]. The association between PTSD and

**Table 5. Risk of bias as determined by the modified Newcastle Ottawa Scale for cross-sectional studies.**

| Studies | Selection | Comparability | Outcome |
|---|---|---|---|
| Chandra P. S. et. Al., 2009 [18] | Moderate | High | Moderate |
| Chowdhary N., Patel V., 2008* [19] | Low | Low | Low |
| Indu P. et. Al., 2018 [20] | Moderate | High | Moderate |
| Jain S. et. Al., 2017* [21] | High | Low | Low |
| Kamimura A. et. Al., 2014 [22] | Moderate | Low | Moderate |
| Sheeba B. et. Al., 2019 [23] | Low | Low | Moderate |
| Supraja et. Al., 2016 [24] | Moderate | Low | Moderate |
| Tichy L.L. et. Al., 2009 [25] | Moderate | High | Low |
| Varma D. et. Al., 2007 [26] | Low | High | Moderate |

*Good quality studies as determined by the NIH Quality Assessment scale

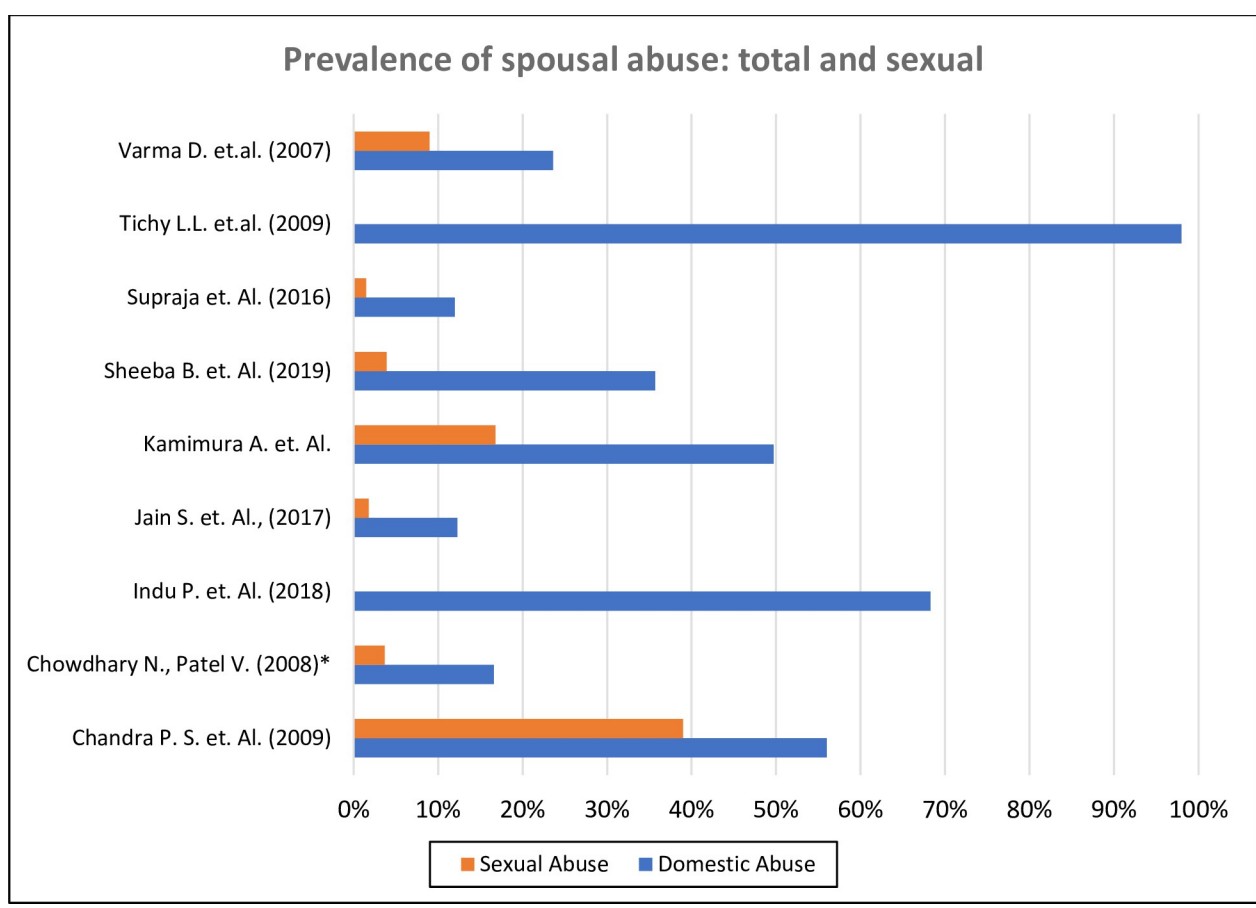

**Fig 3. Prevalence of spousal abuse in each of the included studies; all forms of abuse and sexual abuse.** *Population-level studies representative of the diverse culture of India.

depression symptoms was also significant (r = 0.50) [18]. It was seen that although only 14% of the participants were diagnosed with PTSD, a greater number were exhibiting symptoms which might indicate that subthreshold PTSD is more common [18].

**Secondary outcomes.** Only 1 study of the 2 that that assessed suicidality as one of the outcomes rated as good quality as per the NIH Quality Assessment Scale. Of these studies, one of

**Table 6. Mental health outcomes in each study.**

| Studies | Depression | PTSD | Suicidality |
|---|---|---|---|
| Chandra P. S. et. Al. (2009) | Yes | Yes | No |
| Chowdhary N., Patel V. (2008) * | Yes | No | Yes |
| Indu P. et. Al. (2018) | Yes | No | No |
| Jain S. et. Al., (2017) * | Yes | No | No |
| Kamimura A. et. Al. | Yes | No | No |
| Sheeba B. et. Al. (2019) | Yes | No | No |
| Supraja et. Al. (2016) | Yes | No | Yes |
| Tichy L.L. et.al. (2009) | No | Yes | No |
| Varma D. et.al. (2007) | Yes | Yes | No |

*Good quality studies as determined by the NIH Quality Assessment scale

them observed a significant association between sexual violence by spouse and suicidal behavior after adjusting for confounders but only in the cross-sectional study [19]. This association was not significant in the longitudinal analysis [19]. Suicidal behavior (ideation and attempts) was prevalent among women who reported spousal sexual abuse, where one study estimated a significant, moderate association between suicidality and domestic violence [25].

The qualitative studies included in this review studied the prevalence of different types of domestic violence among the participants and the association between domestic abuse and poor mental health outcomes [27, 28]. One study assessed the prevalence of depression, PTSD, and suicidality [27]. It reported that most women suffered severe sexual abuse which was perpetuated by their spouse and exhibited symptoms of depression, PTSD and suicidal ideation [27]. The other study assessed "Idioms of Distress" (IOD) listed in the DSM-V criteria as the outcome of interest. 6 survivors of domestic sexual abuse were included in the study and all of them displayed symptoms of severe mental distress, anxiety, suicidal thoughts, distrust of people, sleeping and concentration problems, low self-esteem, changed goals and aspirations and feelings of sadness [28]. Participants reported that such feelings and loss of interest in any chores or daily tasks prompted further violence [28]. However, the authors did not study the association of different forms of domestic abuse and the observed IODs [28].

## Discussion

### Review findings

This systematic review identified a limited body of research looking at the mental health implications of spousal sexual abuse, and several gaps in this literature. Although most of the studies did not exclusively look at marital rape and its impact on the victims, they demonstrated associations between spousal violence, and depression and PTSD. This review sheds light on possible adverse mental health outcomes, with depression being the most studied one. Even though several studies combined types of spousal violence into one variable or did not define sexual abuse comprehensively, there is evidence to suggest that spousal sexual abuse is detrimental to the mental health or married women. Almost all the studies that presented depression as an outcome, found it to be significantly associated with spousal IPV [19, 22–24, 26] PTSD was less commonly studied, followed by suicidality and psychiatric distress. One study in this review looked at the association between help-seeking and PTSD, and reported that women who recognized domestic abuse as a common malady and sought help tended to suffer from chronic PTSD and severe symptomology [25]. Other included studies found the expression of PTSD symptoms among women who were victims of spousal sexual abuse, including sexual assault during pregnancy and reported varied outcomes regarding the association [18, 26]. This could be due to the lack of standardization and differences in sampling; for example one recruited women visiting antenatal clinics while another recruited participants by means of newspaper advertisements [18, 26]. The majority of included studies did not observe a statistically significant association between marital rape and suicidality or psychiatric distress, but there could be several reasons for that such as underreporting of symptoms, clinical diagnosis not available (subthreshold), single variable for spousal sexual abuse among others [24, 25, 32].

Qualitative research sheds light on sexual abuse among married couples that was not completely captured by quantitative studies. Majority of the participants suffered from depressive and PTSD symptoms, poor quality of life, loss of interest and will to perform daily activities and even suicidal ideation [27, 28]. The studies also shed light on the lack of awareness about sexual abuse and the normalization of domestic violence. The structured interviews conducted as a part of two of the included studies showed that women were forced to perform sexual acts such as watching and enacting pornography, touching or exposing genitals, forced

anal and oral sex, and even solicitation by spouse [27, 28]. Incest also emerged as an issue but was hardly reported [27]. Thus, in-depth qualitative research can uncover sexual abuse that is not covered by questionnaires and even acts that the victims themselves fail to classify as spousal sexual abuse. Most validated and widely used tools to assess domestic violence only ask about forced penetration or sexual coercion, while leaving out acts such as anal and oral sex, forced masturbation of self or the perpetuator, genital mutation and many others that would also qualify as sexual abuse [27]. This allows for gaps in determination of the extent of sexual violence prevalent among participants.

There can be several reasons for the low levels of reported sexual violence: underreporting due to stigma, lack of awareness of what qualifies as spousal sexual abuses, and rejection of such complaints by society [25, 27]. Indian society provides limited autonomy to women, treating them as property of their spouses [34, 35], a structural relation widely used to justify spousal sexual abuse. Lack of appropriate legal measures that may offer protection to victims of spousal sexual abuse, and secondary victimization by medical personnel act as a major deterrents to reporting [36–38]. Severe sexual abuse negatively impacts help-seeking behaviour, as one of several key factors that limit help seeking, including education status, age, decision-making capacity, source of livelihood and support of family and peers [39].

## International and national relevance

Beyond these findings from India, there is evidence from other LMICs that shows that an association between IPV and adverse mental health outcomes exists. The WorldSafe study that was conducted in four countries (Chile, India, Philippines, and Egypt) found that suicidality and depression are highly common among victims of spousal abuse [19, 40]. After analysis of national level data from the National Health and Family Survey (NFHS), studies revealed that women who report experiencing spousal sexual abuse were more likely to report mental health symptoms rather than a condition itself [33]. This shows the existing stigma around mental health in the Indian society, and rejection of the idea of mental wellbeing being equally important as physical health. Symptoms of depression are often misclassified by survivors as "tension" which can be another factor that deters them from seeking medical help [22, 27]. Domestic abuse was found to be associated with subthreshold and clinically diagnosable depression among married and even pregnant women, indicating the need for screening for violence victimization at hospitals and antenatal clinics [19, 21–23, 26].

## Limitations

This review has some limitations. First, we only searched three databases for articles that fit this criterion. Studies that were not conducted in India were excluded, which could have resulted in missing the abuse within the Indian community living in other countries. However, this does not significantly impact the results of this review as the aim was to explore the lack of research on marital rape in India and address the stigma surrounding this issue. Second, differences in use of standardized tools in both measuring spousal abuse and mental health outcomes can produce differences in observed results. Some studies use nationally collected survey data while others use much smaller single clinic samples. Nonetheless, these different settings give us a strong foundation to build further research on. Sexual abuse within marriage is a very stigmatized subject and research in this field is limited. Considering that both sexual violence and mental health are very sensitive and difficult topics, along with being culturally rejected makes it harder to study the relation between them. But this limitation also reinforces the need for further study in the relation between marital rape and mental wellbeing.

### Strengths and future research

Notwithstanding these limitations, this review demonstrates the importance of elevating martial rape in India as a critical public health concern. The low number of cases reported, and lack of dialogue around marital rape helps explain why most affected women do not seek help for mental disorders. Almost all reviewed studies showed an association between marital rape and adverse mental health outcomes, despite likely widespread underestimation of marital sexual abuse prevalence. The qualitative studies provided relevant context regarding the lack of recognition of actions that may be considered abuse and mental health symptoms. Thus, further research in the association between marital sexual abuse and poor health outcomes, especially mental health is needed. Culturally relevant and validated scales to measure marital rape, awareness about body autonomy and marital rights, comprehensive protocols in healthcare institutions to measure sexual IPV and a more robust legal infrastructure are just some ways to help capture more accurate estimates of the prevalence of marital rape in India. This issue warrants greater attention from researchers as well, and the resulting scholarship can lead the conversation of how marital rape is a pressing public health problem. This can drive action at societal, institutional, and interpersonal levels, and help bring this issue to the forefront, encouraging more people to seek help.

### Recommendations

Marital rape violates fundamental human rights and is linked to adverse psychological consequences. Lawmakers should be made aware of these highly adverse effects, as it may motivate criminalization of marital rape in India. Marital rape and its effects on health and wellbeing warrants greater scholarly focus and nuanced, sensitive research to drive informed and evidence-based decision making.

## Supporting information

**S1 Checklist. PRISMA 2020 checklist for conducting a systematic review.**
(DOCX)

**S1 File. Search strategy for PubMed (includes all search terms).**
(DOCX)

## Author Contributions

**Conceptualization:** Nandini Agarwal.

**Data curation:** Nandini Agarwal.

**Formal analysis:** Nandini Agarwal.

**Investigation:** Nandini Agarwal.

**Methodology:** Nandini Agarwal.

**Project administration:** Nandini Agarwal.

**Software:** Nandini Agarwal.

**Supervision:** Salma M. Abdalla, Gregory H. Cohen.

**Validation:** Nandini Agarwal.

**Visualization:** Nandini Agarwal.

**Writing – original draft:** Nandini Agarwal.

**Writing – review & editing:** Nandini Agarwal, Salma M. Abdalla, Gregory H. Cohen.

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
