## [Decision Letter · Decision Letter 0]

20 Jan 2022

PGPH-D-21-00584

Marital Rape and its impact on the Mental Health of women in India: A Systematic Review

Dear Dr. Agarwal,

Thank you for submitting your manuscript to PLOS Global Public Health. After careful consideration, we feel that it has merit but does not fully meet PLOS Global Public Health’s publication criteria as it currently stands. Therefore, we invite you to submit a revised version of the manuscript that addresses the points raised during the review process.

We look forward to receiving your revised manuscript.

Kind regards,

Nafisa Halim

Academic Editor

Journal Requirements:

1. We have noticed that you have uploaded supporting information but you have not included a list of legends.  Please add a full list of legends for all supporting information files (including figures, table and data files) after the references list.

Additional Editor Comments (if provided):

Reviewers' comments:

Reviewer's Responses to Questions

**Comments to the Author**

1. Does this manuscript meet PLOS Global Public Health’s publication criteria? Is the manuscript technically sound, and do the data support the conclusions? The manuscript must describe methodologically and ethically rigorous research with conclusions that are appropriately drawn based on the data presented.

Reviewer #1: Yes

2. Has the statistical analysis been performed appropriately and rigorously?

Reviewer #1: N/A

3. Have the authors made all data underlying the findings in their manuscript fully available (please refer to the Data Availability Statement at the start of the manuscript PDF file)?

Reviewer #1: Yes

4. Is the manuscript presented in an intelligible fashion and written in standard English?

Reviewer #1: Yes

5. Review Comments to the Author

Reviewer #1: This article is a well-written systematic review on a critical psychosocial problem with profound implications on public health and social justice. The authors have outlined the overall scenario of marital rape and associated mental health outcomes in India. However, this study should be revised to strengthen some areas for scholarly clarity and accuracy.

First, some definitions of marital rape should be cited and discussed in the background. This is critical because of the lack of any specific definition that is used in India. It is okay to cite global literature to inform the readers about the definitions used elsewhere, acknowledging a lack of health, social, administrative, and legal definitions in that context.

Second, the timeframe of the literature search should be stated as "We searched literature from X date to Y date" where X date means the beginning of publications, not the date of searching the same. However, "Y date" can be the date of searching for the last time, which shows the time limit of the included literature. If "Y date" has two values, for example, the authors searched once and repeated the same later, that should be stated clearly, without affecting the beginning date (X date of the search timeframe).

Third, the abstract mentions the "Indian sub-continent" to define the geographic focus, whereas the eligibility criteria focus on Indian women only. It is recommended that the authors should use the study population consistently.

Fourth, although estimating the prevalence of marital rape was part of the objective, the authors did not provide details on this topic in the results section. It should be described narratively in one-two paragraphs alongside specific examples or explanations from the included studies. It should be reflected in the abstract as well.

Fifth, the authors may wish to update their tables mentioning key information on the study methods for each recruited study. These should include but are not limited to information on recruitment strategy, sampling strategy, study design, and analytical approach. It will help the readers to understand the strengths and weaknesses of the included studies.

Sixth, the authors may specify and report whether this study focuses on specific mental health outcomes or all mental health outcomes. The included studies have many mental health issues reported in respective samples. In this scenario, the authors may focus on specific disorders recognized in DSM or ICD or include any psychological condition associated with marital rape, making the review more inclusive from an evidence synthesis perspective.

Lastly, the discussion should present some insights or recommendations for future policymaking and practice. Marital rape is a critical social problem that should be addressed through multilevel efforts, and this study can use the synthesized evidence to provide some recommendations that the authors find suitable in the context of India.

6. PLOS authors have the option to publish the peer review history of their article (what does this mean?). If published, this will include your full peer review and any attached files.

**Do you want your identity to be public for this peer review?** For information about this choice, including consent withdrawal, please see our Privacy Policy.

Reviewer #1: **Yes: **Dr. Md Mahbub Hossain

---

## [Decision Letter · Decision Letter 1]

21 May 2022

Marital Rape and its impact on the Mental Health of women in India: A Systematic Review

PGPH-D-21-00584R1

Dear Ms. Agarwal,

We are pleased to inform you that your manuscript 'Marital Rape and its impact on the Mental Health of women in India: A Systematic Review' has been provisionally accepted for publication in PLOS Global Public Health.

Best regards,

Nafisa Halim

Academic Editor

Reviewer Comments (if any, and for reference):

Reviewer's Responses to Questions

**Comments to the Author**

1. If the authors have adequately addressed your comments raised in a previous round of review and you feel that this manuscript is now acceptable for publication, you may indicate that here to bypass the “Comments to the Author” section, enter your conflict of interest statement in the “Confidential to Editor” section, and submit your "Accept" recommendation.

Reviewer #1: All comments have been addressed

2. Does this manuscript meet PLOS Global Public Health’s publication criteria? Is the manuscript technically sound, and do the data support the conclusions? The manuscript must describe methodologically and ethically rigorous research with conclusions that are appropriately drawn based on the data presented.

Reviewer #1: Yes

3. Has the statistical analysis been performed appropriately and rigorously?

Reviewer #1: N/A

4. Have the authors made all data underlying the findings in their manuscript fully available (please refer to the Data Availability Statement at the start of the manuscript PDF file)?

Reviewer #1: Yes

5. Is the manuscript presented in an intelligible fashion and written in standard English?

Reviewer #1: Yes

6. Review Comments to the Author

Reviewer #1: The revised version of the manuscript is well written and it is likely to inform future research and practice in this area of socio-behavioral health in India and similar contexts.

7. PLOS authors have the option to publish the peer review history of their article (what does this mean?). If published, this will include your full peer review and any attached files.

**Do you want your identity to be public for this peer review?** For information about this choice, including consent withdrawal, please see our Privacy Policy.

Reviewer #1: **Yes: **Md Mahbub Hossain
